# Adipose-Derived Stem Cells in the Treatment of Perianal Fistulas in Crohn’s Disease: Rationale, Clinical Results and Perspectives

**DOI:** 10.3390/ijms22189967

**Published:** 2021-09-15

**Authors:** Etienne Buscail, Guillaume Le Cosquer, Fabian Gross, Marine Lebrin, Laetitia Bugarel, Céline Deraison, Nathalie Vergnolle, Barbara Bournet, Cyrielle Gilletta, Louis Buscail

**Affiliations:** 1Department of Surgery, CHU Toulouse-Rangueil and Toulouse University, UPS, 31059 Toulouse, France; ebuscail@me.com; 2IRSD, University of Toulouse, INSERM 1022, INRAe, ENVT, UPS, 31300 Toulouse, France; celine.deraison@inserm.fr (C.D.); nathalie.vergnolle@inserm.fr (N.V.); 3Department of Gastroenterology and Pancreatology, CHU Toulouse-Rangueil and Toulouse University, UPS, 31059 Toulouse, France; lecosquer.guillaume@gmail.com (G.L.C.); bournet.b@chu-toulouse.fr (B.B.); gilletta.c@chu-toulouse.fr (C.G.); 4Centre for Clinical Investigation in Biotherapy, CHU Toulouse-Rangueil and INSERM U1436, 31059 Toulouse, France; gross.f@chu-toulouse.fr (F.G.); lebrin.m@chu-toulouse.fr (M.L.); bugarel.l@chu-toulouse.fr (L.B.)

**Keywords:** perianal fistula, Crohn’s disease, mesenchymal stem cells, adipose-derived stem cells, allogenic stem cells

## Abstract

Between 20 to 25% of Crohn’s disease (CD) patients suffer from perianal fistulas, a marker of disease severity. Seton drainage combined with anti-TNFα can result in closure of the fistula in 70 to 75% of patients. For the remaining 25% of patients there is room for in situ injection of autologous or allogenic mesenchymal stem cells such as adipose-derived stem/stromal cells (ADSCs). ADSCs exert their effects on tissues and effector cells through paracrine phenomena, including the secretome and extracellular vesicles. They display anti-inflammatory, anti-apoptotic, pro-angiogenic, proliferative, and immunomodulatory properties, and a homing within the damaged tissue. They also have immuno-evasive properties allowing a clinical allogeneic approach. Numerous clinical trials have been conducted that demonstrate a complete cure rate of anoperineal fistulas in CD ranging from 46 to 90% of cases after in situ injection of autologous or allogenic ADSCs. A pivotal phase III-controlled trial using allogenic ADSCs (Alofisel^®^) demonstrated that prolonged clinical and radiological remission can be obtained in nearly 60% of cases with a good safety profile. Future studies should be conducted for a better knowledge of the local effect of ADSCs as well as for a standardization in terms of the number of injections and associated procedures.

## 1. Introduction

Crohn’s disease (CD) is a chronic inflammatory disease, characterized by idiopathic transmural inflammation anywhere along the gastrointestinal tract [1]. The increasing incidence and the difficulties of treating some lesions represent real problems for public health and medical management. The fistulizing phenotype of disease is notoriously difficult to treat, especially when located perianally [2,3,4,5,6]. Anal and perianal localizations in CD are frequent and the evolution vary from regression to a prolonged and chronic state. In this latter case the lesions can worsen with large mucocutaneous and muscular involvement that considerably modify the anorectal architecture [3,4,5,6,7]. Moreover, these lesions seem to have an autonomous evolutionary power: a treatment can be perfectly effective on the intestinal location while the anoperineal lesions continue to evolve on their own account. Between 40 to 50% of CD patients suffer from fistulas, and among these 50% have a perianal fistulizing disease [3,4,7,8]. The other localizations are entero-enteric, rectovaginal, entero-cutaneous, entero-abdominal, and entero-vesical. Perineal fistulas in CD can be treated with a variety of surgical procedures and drugs. None of these are perfectly effective and there is always a risk of recurrence. It is therefore essential to preserve the function of the anus as much as possible by avoiding repeated mutilating surgeries which increase the risk of incontinence and permanent stoma. Unfortunately, in about one third of patients with fistulizing perianal CD the disease is refractory. Therefore, fistulizing perianal CD is a marker of disease severity [1,4,8].

The central problem is the treatment of complex fistulas. Seton drainage followed by immunosuppressive drugs has not proven to be effective. Antibiotics (metronidazole and ciprofloxacin) only have a temporary effect on symptoms and their prolonged use can induce adverse side effects (peripheral neuropathy, tendon rupture). In the presence of suppuration, the best conventional approach is surgery with drainage of any abscesses, fistulotomy, or placement of a seton [6,7]. In some patients with severe perianal disease, a proctectomy and a diverting stoma may be required. Previous studies (including our experience) have suggested that seton drainage for several months combined with anti-TNFα can result in closure of the fistula in 70 to 75% of patients [9,10,11,12,13,14,15,16,17].

However, the remaining 25% of patients require repeated procedures which carry a high risk of destructive lesions and functional consequences (risk of incontinence) as well as severe alterations in quality of life [18,19].

In this context, there is room for other procedures such as biological glue, biological collagen or in situ injection of autologous or allogenic mesenchymal stem cells (MSC) from bone marrow (BM-MSC) or adipose tissue (adipose-derived stem/stromal cells—ADSC). Despite the diversity of protocol designs (autologous versus allogenic, number of injected cells, single or repeated injections, addition of glue or not, time of evaluation, etc.) the numerous phase I and II trials are encouraging with regard to both safety and efficacy [20,21,22,23,24,25]. ADSCs have been more widely assessed since they are easier to access and harvest than other MSCs, while maintaining all the crucial properties such as multipotency, self-renewal, anti-inflammatory effects, immuno-modulation and pro-angiogenesis [21,26,27,28,29]. ADSCs can also be differentiated in vitro into adipocytes, chondrocytes, myocytes, osteoblasts, and neurocytes. These properties explain their use in clinical trials for the treatment of various diseases such as diabetes mellitus, liver and cardiac diseases, corneal lesions, articular and cutaneous lesions, or vascular ischemia. In addition, stem cells and particularly ADSCs play a key role in reconstructive or tissue engineering medicine [30]. The aim of the present work is to review the relevant properties and the application of ADSCs in the treatment of fistulizing perianal CD, including published and ongoing clinical trials.

## 2. Treatment of Perianal Fistulizing Crohn’s Disease

A positive diagnosis and classification of Crohn’s perianal fistulas relies in part on a clinical proctology examination, ideally under general anesthesia, and imaging [31,32,33,34,35]. In this case, the anorectal ultrasound in 2D or 3D is particularly interesting. MRI is often recommended as a first-line procedure because of its accuracy, an accuracy that has been evaluated to be between 76 and 100% especially for the assessment of fistulous tracts as simple or complex, for visualization of primary and secondary orifices, for diverticula and abscesses, and finally for assessment of sphincter status [36,37,38]. According to the American Gastroenterological Association’s classification, a simple anal fistula is defined as having a single low superficial trans-sphincteric track with no abscess formation, stenosis, or anorectal inflammation. All other forms are classified as complex and comprise the majority of lesions (high trans-sphincteric, extra-sphincteric, or supra-sphincteric tracks). Complex fistulas are characterized by two different pathogenic processes, including granulomatous inflammation (specific to CD) and suppuration, leading to abscess formation [31,32,33,34,35]. Treatment is always justified in cases of complex fistulas. Before initiating medical treatment, the fistula should be well drained (with no underlying abscesses or infections). The standard treatment is seton drainage alongside fistulotomy, mucosal advancement flap or ligation of the inter-sphincteric fistula tract [39,40,41,42]. In addition, we and others have demonstrated that prolonged seton drainage is more effective than short-term application [17,42]. It has been also demonstrated that the addition of a fistula plug (biological glue, collagen plug …) improves the results of the seton drainage [43,44,45].

Medical management is mainly based on anti-TNF and immunosuppressive treatments such as thiopurines or methotrexate. When administered alone, these generally have a modest effect on the rate of fistulous tract closure [46]. After drainage, the best treatment seems to be anti-TNFα antibodies and good results have been obtained with Infliximab and adalimumab, either alone or in combination with an immunosuppressive drugs treatment [14,15,16,17,47,48,49]. A systematic review and meta-analysis including 27 controlled trials found that TNF antagonists are effective for the induction and maintenance of perianal fistula response and remission [47]. Anti-TNF-alpha drugs achieve a better resolution of drained fistula tracts than placebos, and the initial therapeutic benefit appears to be maintained during maintenance therapy [47]. All these studies suggest that combined surgical drainage and anti-TNFα results in a closure of the fistula (mainly complex) in 50 to 70% of patients [16,17,50,51,52,53,54,55,56,57]. Several reports have described the effect of more recent treatments such as Vedolizumab (anti-α_4_β_7_ integrin antibody) and Ustekinumab (anti-IL-12/23) [58,59,60,61,62,63]. These compounds have been applied in patients suffering from complex fistulas, most of whom had already received anti-TNFα treatments. There is a low success rate with Vedolizumab, but Ustekinumab appears as a potential effective therapeutic option in perianal refractory CD, warranting further prospective studies [58,59,60,61,62,63].

Despite the progress provided by the association of surgical drainage with anti-TNFα treatment, 20 to 25% of patients have a refractory fistulizing disease, sometimes resulting in diverting stoma and more frequently proctectomy. We and others have identified some factors that are associated with these complex situations for patients and doctors: female patients, association with rectal involvement (proctitis, rectal stenosis), and association with rectovaginal fistulas [64,65,66].

There is thus a place for other approaches in combination with the standard treatment. Several teams have tested local application of cyanoacrylate glue or bio prosthetic plugs, but there is serious hope concerning the use of cellular therapy by direct in situ injection of MSCs, principally ADSCs. Despite the initially disappointing results for systemic administration of MSCs, allogenic ADSCs (Cx601, Darvadstrocel, Alofisel^®^) locally injected in perianal fistula tracts have been shown to induce long-lasting beneficial effects and the drug has been recently approved in Europe (European Medicines Agency) [67,68]. However, there are still several issues to be resolved in this field: what is the exact mechanism of action of the ADSCs? What is the best administration protocol in terms of number of cells, timing after drainage and systemic biotherapy, and associated procedures (glue, plug or not)? What is the exact risk of a potential allogenic process?

## 3. Properties of ADSCs and Their Clinical Application

### 3.1. Origin and General Properties of MSCs

Within the bone marrow, mesenchymal precursors have been shown to differentiate in vitro not only into osteoblasts, but also into chondrocytes and adipocytes [69,70,71,72,73,74,75]. MSCs have also been identified in different tissues. However, the process to obtain MSCs from tissues is more complex than that of BM-MSCs. There is indeed a need to digest the tissue with proteolytic enzymes (Figure 1). Of the various tissues studied, many teams have focused on adipose tissue because it is easy to collect in large quantities by liposuction. These cells are known as ADSCs for adipose-derived stroma/stem cells. The frequency of ADSCs in adipose tissue is 100 to 500 times higher than that of BM-MSCs in bone marrow [69,70,71,72,73,74,75]. Although similar in many respects, there are several differences between these two types of cells in terms of immunophenotype, differentiation potential, transcriptome, proteome, and immunomodulatory activity. ADSCs appear for instance to have superior angiogenic potential and anti-inflammatory effects than BM-MSCs [74,75].

The commonly accepted definition of an adult stem cell population is based on three key criteria: the cell population must be (1) immature, (2) capable of regenerating its tissue of origin and therefore have high proliferative potential and be generally multipotent, and (3) capable of self-renewal [76,77].

Multipotency is a widely accepted property of MSCs since these cells are, by definition, capable of differentiating in vitro into adipocytes, osteoblasts, and chondrocytes [26,72,73]. In addition, many papers indicate that MSCs are also capable of generating vascular smooth muscle cells [26,72,73]. This multipotency means that these cells have immense potential, making it possible to envisage tissue reconstruction, especially as some studies report a broader differentiation potential, with a profile similar to that of embryonic stem cells: differentiation into endothelial cells, skeletal and cardiac muscle cells, neural cells, hepatocytes, and epithelial cells [21,27]. In addition to tissue reconstitution by repopulation, reprogramming into intestinal, pulmonary or neural cells can also be proposed [21,27,28,78,79,80].

Self-renewal is defined as the ability of a stem cell to give, after mitosis, at least one cell identical to itself, which retains its full potential for differentiation and proliferation.

Plasticity is the ability of mesenchymal cells to move from one differentiation state to another under the influence of extracellular factors. For example, chondrocytes in the intermediate plate hypertrophy before transforming into osteoblasts. MSC clones differentiated into adipocytes, osteoblasts, chondrocytes or vessels may give cells of alternative mesenchymal lineage after modification of the culture conditions [27,29].

In general, the initial definition given for an MSC was as follows: plastic adherence in standard culture conditions and in vitro proliferation, surface marker expression of CD105, CD73 and CD90, lack of expression of CD45, CD34, CD14, CD11b, CD79, or CD19 and HLA-DR, multipotent differentiation potential into diversified cell types derived from the mesoderm under specific stimulus in vitro (osteoblasts, adipocytes, and chondrocytes). Recently, the nomination of MSCs as “medicine signaling cells” has been proposed due to their wide range of biological activities and clinical applications [77,81].

### 3.2. Isolating ADSCs

Human bone marrow MSCs are obtained by a very simple process. After bone marrow puncture, the cells are suspended and seeded in culture dishes in a liquid medium with added serum (human or bovine), then selected to obtain optimal growth of the adherent cells [82].

ADSCs are obtained by liposuction, usually abdominal. Then comes a digestion step which is crucial to free the cells from the extracellular matrix. However, this step induces great differences in the obtained cellular populations depending on the protocol and the proteolytic enzymes used. It is important to keep in mind that the isolation of distinct cell populations due to the use of different protocols can explain the variable experimental results, a point that is often overlooked. After digestion, the lipid-filled adipocytes floating in the cell suspension are removed from the so-called stromal vascular fraction (SVF) of the tissue remaining in the centrifuged pellet. This fraction corresponds to a heterogeneous cell population that contains, among others, hematopoietic cells, endothelial cells, and immature cells [25,27,72,73]. The culture of this fraction and the medium used allow mesenchymal-type cells to adhere and thus be selected by washing, as in cultures established from BM [27,28,80,81]. This is a selection by adhesion and expansion in a specific culture medium allowing the survival and amplification of ADSCs. However, such a procedure does not result in purification as could be expected from cell sorting by flow cytometry. A large number of culture conditions have been developed based on the use of several different media, growth factors, serum or platelet lysates, number of passages, and culture times [83,84,85,86,87,88,89]. These different conditions for generating the cells result in a heterogeneity of the product obtained for use in clinical trials. This latter point provides several difficulties for comparison of results between clinical studies.

Figure 1 gives a schematic description of the different steps required to obtain ADSCs. Recently, it has been demonstrated that by using specific inducers for cell cultures in the laboratory, these ADSCs have the ability to engage differentiation into the cellular line needed [83].

### 3.3. Properties and Molecular Aspects of MSCs

MSCs, and more particularly ADSCs, exert their effects on tissues and effector cells through paracrine phenomena, including the secretome and extracellular vesicles [21,25,27,28,29,90,91,92,93,94,95,96,97,98,99]. The secretome and its role have been well studied through experiments conducted in vitro with conditioned media transfers, in particular by ADSC cultures [95,96,99]. Although these effects vary according to the seeding conditions, the culture time, and the medium itself, it is evident that MSCs secrete bioactive molecules (interleukins, growth factors …) which act in contact with the host tissues and cells [21,28,91,97]. A second mode of action is the secretion of extracellular vesicles [90,91,92,94]. These are made up of exosomes, microvesicles, and apoptotic bodies [27,28,29]. They also carry a number of potentially active principles such as membrane and cytosolic proteins, transcription factors, DNA, mRNA, rRNA, miRNA, and various key transduction signal molecules [90,99].

The anti-inflammatory properties are mainly due to the production/secretion of cytokines such as interleukine-4, interleukine-10, transforming growth factor, chemokine-(C-C motif)-ligand 18, but also hepatocyte growth factor (HGF), indoleamine 2,3-dioxygenase (IDO), prostaglandin E2 (PGE2), nitric oxide (NO), and Lipoxin A4 [21,27,28,29,100].

The anti-apoptotic, pro-angiogenic proliferation and cell migration properties are due to the secretion of growth and pro-angiogenic factors such as vascular endothelial growth factor (VEGF), insulin growth factor (IGF), HGF, nerve growth factor (NGF), Neurotrophin-3 and also to the mitochondrial transfer by microvesicles [101,102,103].

Immunomodulatory and even immunosuppressive properties have been validated in vitro and in vivo. These effects are due to the MSCs secreting multiple molecules such as cytokines that include interleukin-13 (IL-13), transforming growth factor (TGF-α), and HGF, and enzymes such as IDO, prostaglandins (notably PGE2), and NO. In general, the T lymphocyte function/proliferation is inhibited and the regulatory T cell function (especially CD4 + T-reg) is promoted [21,91,104,105,106,107]. Figure 2 shows the main mechanisms of action of ADSCs both in vitro and in vivo.

Other properties are also important for their clinical application. They have an immuno-evasive status due to the absence of expressions of class II major histocompatibility complex (MHC) and B7-1 co-stimulatory molecules and a low expression of class I MHC [104,106].

Some other properties have also been mentioned such as homing within the damaged tissue, anti-oxidant, anti-bacterial, anti-viral, and anti-tumor effects [108,109,110,111].

### 3.4. Advantages of ADSCs versus BM-MSCs

The immunophenotypes of MSCs and ADSCs are 90% identical and the existing minor differences can explain their different efficacy. Various pre-clinical studies have compared the in vitro effects of ADSCs with BM-MSCs. These studies established that ADSCs have a higher and more prolonged replication rate in culture, associated with a longer morphological and genetic stability (regardless of the age of the donor). They also have a greater anti-inflammatory and anti-angiogenic potential [74,75,112,113,114,115,116,117]. Their immuno-modulatory or even immuno-suppressive effect seems to be similar to that of BM-MSCs, however they have a more marked effect on the suppression of IgG production, the differentiation of monocytes into dendritic cells, and the inhibition of peripheral blood mononuclear cell (PBMC) proliferation (experiments on ADSC and PBMC co-cultures versus PBMC and BM-MSC co-cultures). Furthermore, in co-cultures of ADSCs and monocytes, the production of IL-10 is higher while that of IL-6 is lower (compared with co-cultures using BM-MSCs) [75,112,117]. The latter effects lead to a more pronounced inhibition of dendritic cells. ADSCs express even less HLA class-1 molecules compared with BM-MSCs (thus ADSCs are more immune-evasive) [117]. Last but not least, it is easier to access adipose tissue and the amount of stem cells in adipose tissue is higher than in BM samples [28,29].

### 3.5. Autologous versus Allogenic ADSCs

In theory, autologous cells are more suitable for any kind of cell therapy. Indeed, they are the ideal choice for obvious histocompatibility reasons. Nevertheless, certain constraints can be identified, including the need to standardize the collection method and ex vivo culture protocol in order to obtain cells that are similar from one patient to another in terms of both number and properties. It is also necessary to systematically avoid any infection during the collection, transport, and culture process. Furthermore, in CD, the prescription of immunosuppressive drugs may alter the intrinsic properties of adipose tissue, although most studies do not seem to support this. The patients’ age and the presence of co-morbidities associated with their disease can also be responsible for an alteration in the quality of their autologous cells.

The use of SVFs shortens the process of developing ADSCs, but the resulting purity and standardization remain questionable. Indeed, SVFs are still a mixture of cells and their reproducibility in terms of ADSC richness and properties has not been fully established to our knowledge. Finally, to obtain autologous ADSCs, the need of a surgical liposuction is an additional invasive procedure in patients that have already undergone surgery for fistula in the context of an inflammation state and a sometimes-precarious nutritional status. It is for most of these reasons that the production of allogeneic ADSC cells has been developed, with the advantages of traceability in terms of sterility, maintenance of properties, and standardization of cell numbers and batches [118]. Although long-term follow-up studies on the allogeneic response produced in recipients are still awaited, the fact that most ADSCs do not express class II MHC and B7-1 co-stimulatory molecules (and express very little class I MHC) may favor the absence of immediate rejection or secondary immunization against allogeneic therapeutic cells [21,27,28,115,117]. The allogeneic ADSC approach can therefore be easily envisaged, with the constitution of a frozen bank from a “universal donor”, contrary to BM-MSCs. In addition, a batch can be used for a larger number of patients instead of a preparation “patient by patient”. This is a distinct advantage when developing an industrial process compatible with obtaining a marketing authorization. A recent study appears to demonstrate this (conducted in patients with CD fistulas applying Alofisel^®^), but it must be confirmed if repeated injections or re-treatment in case of recurrence are to be planned in the future. In addition, when considering the specific case of CD, we really don’t know whether autologous ADSCs “fix” and modulate the immune response to a greater extent than ADSCs from a healthy donor [119].

### 3.6. Clinical Applications and Safety of ADSCs

MSCs have been applied to a wide range of pathologies due to their pluripotency and multiple anti-inflammatory, anti-apoptotic, antioxidant, immunomodulatory, and angiogenic effects with a production of growth factors and cytokines [21,27,28,29]. A non-exhaustive list includes cardio-vascular diseases, diabetes, inflammatory intestinal and rheumatological diseases, and autoimmune diseases such as lupus and multiple sclerosis [120,121,122,123,124,125,126,127]. Their action of homing to damaged tissue and regenerative properties are at the heart of regenerative medicine, with applications in cardiovascular (ischemia), hematological (medullary regeneration), hepatic (cirrhosis) and wound healing pathologies in the broadest sense [121,128,129]. More recently, they have been proposed for respiratory distress syndrome, in particular that induced by SARS-COV2, for their anti-inflammatory and immunoregulatory properties [130]. One of the advantages of ADSCs is that they are easier to access and harvest by means such as subcutaneous liposuction, which is a much less painful procedure than harvesting bone marrow stem cells. Additionally, their use is less subject to ethical controversies because they are harvested from adult fat, and not embryonic or fetal cells. In terms of short- and medium-term safety (up to 3 years of follow-up), the profile is excellent if we combine the results of more than 200 trials using ADSCs for all indications [30,131]. More important, the first follow-up studies of patients after in situ injections of allogenic ADSCs are encouraging in terms of an absence of allogenic immunization [119]. Among all these clinical trials, no evidence of a potential pro-oncogenic role of ADSCs has been shown despite pre-clinical studies that raised some evidence that adipose stem cells might have a role in the tumor microenvironment as well as in tumor promotion [132,133,134]. A recent report demonstrated that human lipoaspirate material had no pro-oncogenic properties in vitro [135]. However, long term in vivo experiments and follow-up of treated patients are still needed.

In the context of CD anoperineal fistulas, the main benefits lie in their anti-inflammatory, anti-angiogenic, immunomodulatory, and regenerative properties [28,136,137,138]. Their anti-infectious effects may also be beneficial but are yet to be proven. Similarly, the relationship with ambient mucosal inflammation (proctitis and rectitis) and the intestinal microbiota is unknown or badly understood.

In addition, the molecular mechanisms underlying the positive effects of ADSCs in the specific case of CD ano-perineal fistulas remains to be extensively deciphered. A work from Aso K. et al. has identified a role for interleukin-4 and interferon-gamma in the protective effects of ADSCs in an in vivo model of aged mice in which gastrointestinal mucosal functions were impaired [139]. To our knowledge this is the only study that has explored molecular mechanisms. Most of the other studies have focused on cellular functions as described in the proposed Figure 3 recapitulating the known mechanisms of ADSCs-induced protection against Crohn’s disease fistula.

In a model of colon fistula developed in rats the local application of ADSCs resulted in a significant rate of fistula closure when compared to placebo [136] but the mucosa was not inflamed. A specific and reliable model of ano-perineal fistula has been recently developed in rats bearing rectitis [140]. It could be used to more precisely characterize the cellular and molecular effects of ADSCs in CD fistulas. Finally, some evidence exists in vitro and in vivo for the role of ADSCs in wound healing through the increase of cell proliferation, invasion, angiogenesis, and neovascularization [28,29].

## 4. Clinical Trials Using ADSCs in Perianal Fistulas in Crohn’s Disease

### 4.1. Published Papers

Most of the clinical trials are early phase (I and II) and only one phase III trial has been completed. This involved allogeneic ADSCs developed by the Takeda^®^ company (Darvadstrocel, Alofisel^®^) and resulted in a marketing authorization in Europe. Table 1 lists the main published studies that included more than 10 patients with anoperineal fistulas in CD (mostly refractory) and who were injected with ADSCs in situ [67,68,141,142,143,144,145,146,147,148,149,150,151]. Most of the patients concomitantly received TNFα antagonists and/or immunosuppressants. Figure 4 details the different types of complex anoperineal fistulas in CD as well as the different means of injecting ADSCs in these cases. The analysis of these trials shows a great disparity in terms of study design (single or multicenter, controlled or not …), the type of cells and preparations used (autologous, allogeneic, SVF), the number of cells injected (from 1 to 12 × 10^7^), the mode of injection (intra and/or peri-fistula, one or more injections …), the associated procedures (setons, biological glue, plug), the means (clinical only, clinical and MRI) and timing (4 to 48 weeks) of the evaluation or follow-up, and concomitant systemic treatments for CD [67,68,141,142,143,144,145,146,147,148,149,150,151]. Most importantly, in all these trials, the problem lies in defining the response, since this varies in terms of clinical response, clinical remission, combined clinical and radiological response, and complete or incomplete closure. Nevertheless, it should be noted that in studies with a limited number of patients, the complete cure rate ranged from 46 to 90% [67,68,141,142,143,144,145,146,147,148,149,150]. In the sole controlled phase III study, complete healing of fistulas was significantly higher compared with the placebo group. In addition, serious adverse events due to the in situ injection of ADSCs were rare. It should also be noted that most studies included complex anoperineal fistulas, but only one study also included anovaginal fistulas [149].

Despite all these discrepancies, the results of the controlled phase III study give hope and place ADSC injection in the therapeutic arsenal for the treatment of anoperineal fistulas in CD [67,68]. However, the new biotherapies have yet to provide a significant improvement in the healing of these lesions which are still refractory to several therapeutic lines in association with classic drainage surgery. This pivotal phase III trial (ADMIRE-CD) can be resumed as follows. It was conducted with allogeneic ADSCs in a phase III, randomized, double-blind, placebo-controlled design. One hundred seven CD patients were treated with allogeneic ADSCs versus 105 with saline (single injection into and around the fistula) for anoperineal fistulas that were initially treated with curettage and seton placement. Note that only patients with absent or moderate intra-luminal disease (CDAI score ≤ 220) were included and with no active proctitis, rectal stricture, or rectovaginal fistula or stoma. The assessment was blinded for the gastroenterologists and radiologists for the clinical and MRI examinations respectively. The primary endpoint was combined remission, i.e., closure of all treated external fistulous orifices and absence of a collection >2 cm at MRI. A secondary endpoint was clinical remission alone with 100% closure of external orifices. Two publications report the results at 24 and 52 weeks respectively [67,68]. Combined remission was observed after ADSC injection in 50% and 56.3% of cases after 24 and 52 weeks respectively (versus 34% and 38.6% for the placebo *p* = 0.024 − 0.010). Clinical remission alone was observed in 57% and 59.2% after 24 and 52 weeks respectively, compared with 41% and 41.6% for the placebo (*p* = 0.064 − 0.013).

The side effects were reported to be similar in the two groups, mainly anal abscesses and proctalgia. The authors do not suggest that the ADSCs alone were responsible for these side effects, but that the entire procedure, including surgery and injection of cells or saline, was the probable cause. In general, the other trials (which admittedly included only a few patients) did not report any severe side effects due to the injection of the ADSCs themselves. From 0 to 30% of patients (mean around 15%) experienced a side effect, mainly proctalgia, abscesses, or fistula [30,67,68,131,141,142,143,144,145,146,147,148,149,150]. Some other studies have been published in which BM-MSCs (in situ injection) were evaluated in the treatment of refractory anoperineal fistulas in CD using various trial designs. To our knowledge, no study has clearly compared the effects of ADSCs with those of BM-MSCs in this indication. ADSCs however do seem to induce a similar percentage of complete healing, but once again the number of patients included was never more than twenty five [20,21,22,23,24,25,137,138,152,153]. Even if long-term studies are still lacking but are now in progress (see below ongoing trials), local MSC injection for perianal fistula in CD continue to support long-term efficacy while maintaining a favorable safety profile. The evidence for systemic MSC infusion in luminal CD remains mixed due to marked methodological heterogeneity and unclear safety profiles [154].

In addition to CD, cryptoglandular fistulas can also be difficult to treat. ADSCs have also been tested for this indication in phase I and II trials with similar results to those obtained in CD [155,156]. Finally, although this is a different issue to anoperineal fistulas but equally crucial, ADSCs have also been evaluated in refractory rectovaginal fistulas in CD. The size of the fistula can be reduced, but complete closure is difficult [157,158]. Further studies with larger numbers of patients are needed.

On the whole, the success of allogenic cells greatly facilitates clinical progress as it can use one or more universal donors and avoids additional surgical procedures in the patient (whether BM-MSC or ADSC) and otherwise not fully standardized ex vivo cultures. It is noteworthy that in the published studies (Table 1) the CD activity index (CDAI) was generally below 200, characterizing moderate to active disease. Therefore, the concomitant systemic treatment for CD is important to reduce luminal inflammation while cell therapy is applied to treat a locally refractory disease.

### 4.2. Ongoing Trials

The main ongoing trials applying ADSCs in the treatment of CD anoperineal fistulas are listed in Table 2. We lack some details as to the number of cells injected, but the trials are divided into autologous, allogeneic, and SVF-derived ADSCs. In the case of the use of SVF, the preparation involves extemporaneous preparation devices (Cytori Celusion System^®^ or Revolve System Acelity^®^) and/or combination with an associated injection of raw adipose tissue. The number of patients is small, except in the extension of the original ADMIRE study, which is being conducted in the USA in combination with a safety and side effect follow-up. Repeated injections are also being evaluated with this same therapeutic principle. In parallel, trials using BM-MSCs are underway with an extension to the treatment of ileal anal anastomosis and ileal pouch fistulas and including immune-monitoring (for instance screening for HLA Class I & II antibodies).

## 5. Perspectives and Future Developments

We can conclude that injection of autologous or allogenic ADSCs, following appropriate surgical and medical preparation, represents a promising combination strategy for the management of resistant perianal fistulas. However, multiple prospects exist for the treatment of CD anoperineal fistulas by cell therapy. The aim is to standardize the trials in terms of inclusion criteria, route of administration, type (autologous, allogeneic or SVF) and number of cells, mode and number of injections, associated surgical procedures, clinical and radiological evaluation criteria, and immuno-monitoring. The relative robustness of the trials of allogeneic ADSCs in terms of patient numbers and follow-up opens the door to using allogeneic stem cells in this indication and permits a standardized and traceable preparation of cells from healthy donors.

However, some questions are yet to be answered. First of all, what is the definition of a refractory anoperineal fistula? Indeed, the factors of resistance to “classic” treatments are well known, such as the presence of active rectitis and proctitis, the existence of a rectovaginal fistula, gender, and the multiple and complex nature of the fistulas. It is therefore still necessary to evaluate the ideal profile of patients who can benefit from ADSC injections.

On the other hand, if a fairly simple surgical treatment with placement of a seton(s) is a good prerequisite, problems will arise concerning the “timing” of the ADSC injection, the mode of injection (external and/or internal intra-fistula, peri-fistula injection) and the associated procedures such as the “placement of biological glue”, for example. Once again, treating the luminal inflammation by systemic treatments is crucial as the cell therapy approach is devoted to a local refractory disease, mainly complex ano-perineal fistulas. In Figure 5, an algorithm for the treatment of anoperineal suppuration and complex fistula in CD is proposed according to already published national and international recommendations [159,160]. This algorithm includes stem cell therapy in the cases of persistent complex fistula without active luminal disease and after a well conducted surgical and medical treatment including TNFα antagonist.

At the moment there are not enough scientific data allowing the formation of unequivocal opinion regarding the systematic use of stem cells in the treatment of perianal CD. It is important to carry out ancillary research by studying, for example, the behavior of the rectal mucosa (before and after the ADSCs injection), as well as the evolution of the microbiota, mucosal and circulating inflammatory markers. In addition, there is some unresolved questions especially on the viability and potential cell death and loss of activity with local administration. To our knowledge only one experimental study conducted in rats and using a bioluminescent marker, demonstrated that the presence of cells decreased rapidly after two days post-injection within a fistula [132]. This poses the problem of repeated injections of ADSCs but the rhythm remains to be fixed. In addition, biocompatibility studies should also be conducted to evaluate the injection system to ensure that the number of cells put in the syringe is really delivered to the injection site. This number strongly depends on the dead volume of the injecting system and the nature of the material to ensure that the right number of cells is really delivered to the injection site(s).

Another field of investigation has been opened up concerning the conditioning of ADSCs [82,161,162], with the possibility of orienting the differentiation and secretory profile in vitro. Nevertheless, it is also important to understand the mechanisms by which ADSCs manage to close CD fistulas: is it by tissue regeneration and healing effect or by anti-inflammatory and immuno-modulation effect? What is the exact role of the secretome and extracellular particles? There are some other limitations such as the fact that the behavior and secretions of rodent ADSCs are not identical to human ADSCs.

We feel that most of this future pre-clinical and clinical research will be a serious help for shedding light on some grey areas regarding the mode of action of ADSCs, the exact protocol to be followed for treating fistulas and their real place in the management of perianal fistulas in Crohn’s disease.

## Figures and Tables

**Figure 1 ijms-22-09967-f001:**
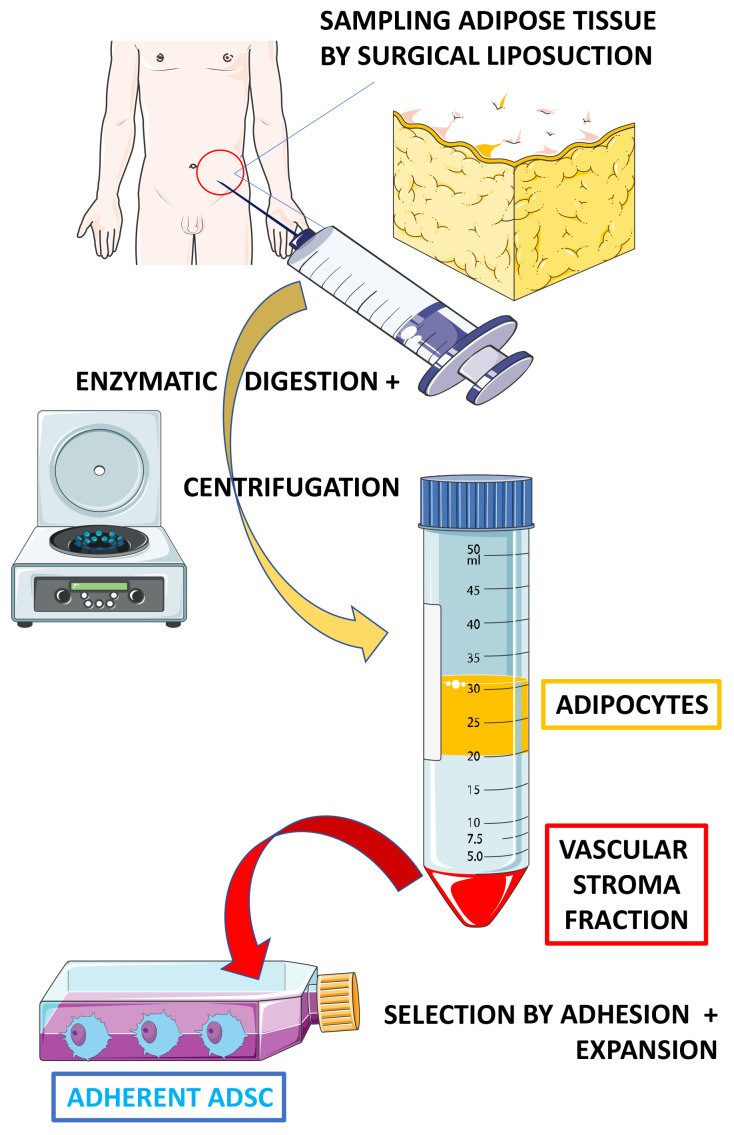
Method for isolating adipose-derived stem cells (ADSC). The adipose tissue is minced with fine scissors or a scalpel, then washed, treated with collagenase and centrifuged. The resulting pellet (serum vascular fraction of hematopoietic, endothelial, vascular smooth muscle, fibroblastic and immature cells, AND ADSCs) is then resuspended in a complete medium before seeding in a culture plate and cultured in a specific medium that facilitates ADSC selection.

**Figure 2 ijms-22-09967-f002:**
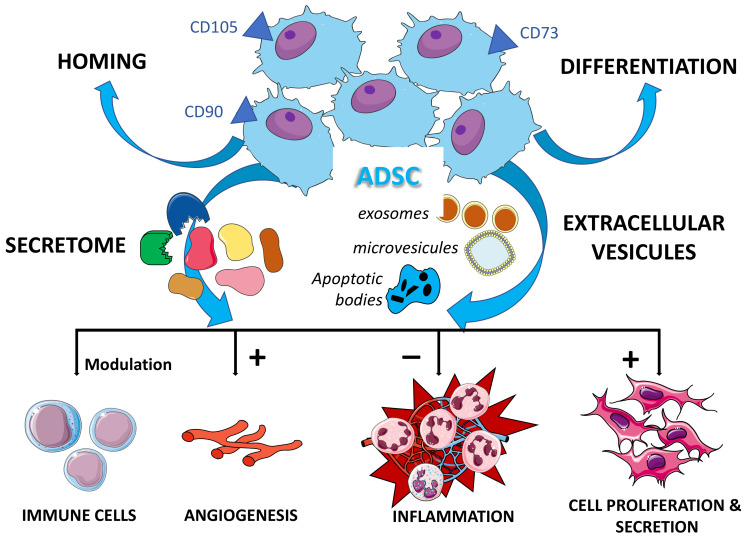
Main properties of adipose-derived stem cells (ADSC). (The secretome mainly contains interleukins, growth factors, indoleamine 2,3-dioxygenase, prostaglandin E2, NO, Lipoxin A4, Neurotrophin-3, etc.; the extracellular vesicles carry membrane and cytosolic proteins, transcription factors, DNA, mRNA, rRNA, miRNA and key molecules of various transduction signals …).

**Figure 3 ijms-22-09967-f003:**
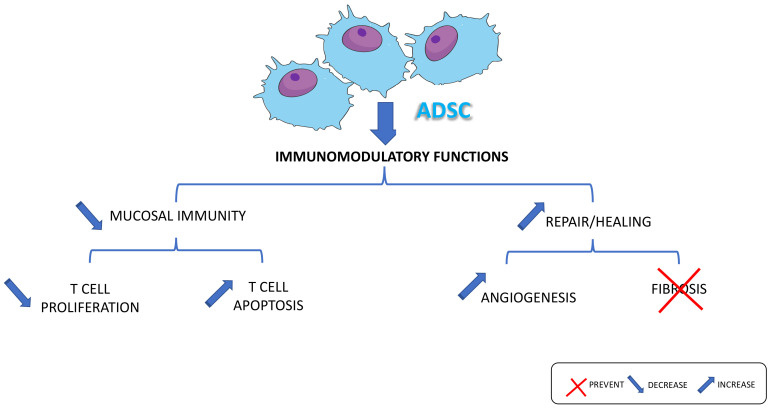
Mechanisms involved in adipose-derived stem cells (ADSCs)-associated beneficial effects in the inflamed gut.

**Figure 4 ijms-22-09967-f004:**
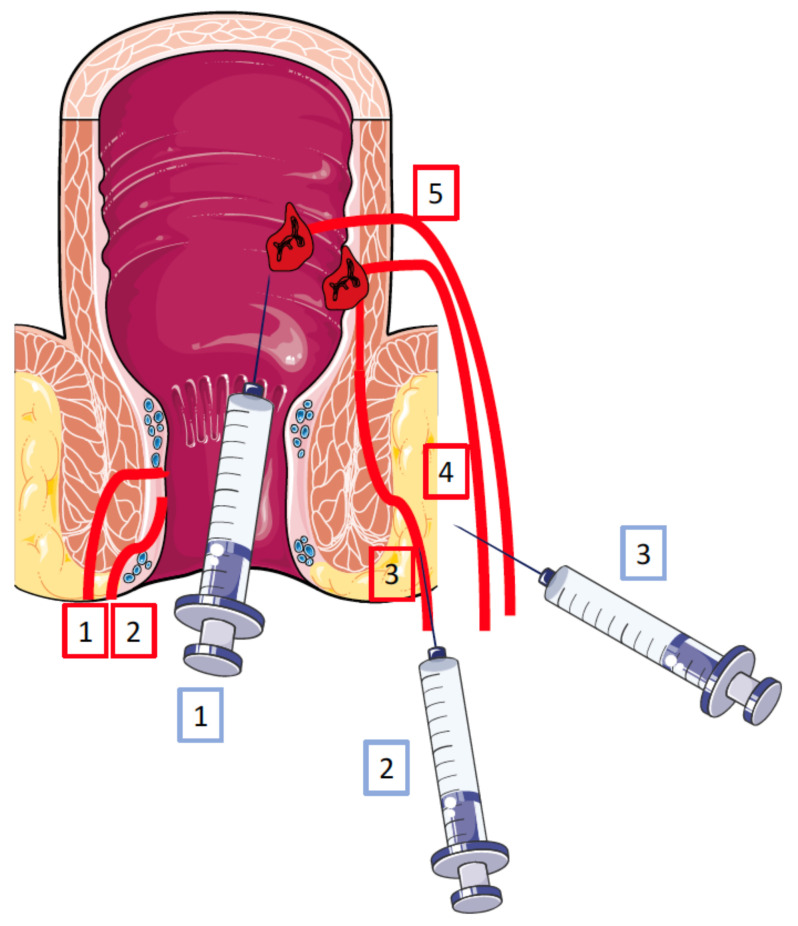
Schematic representation of anoperineal fistulas in Crohn’s disease and of adipose-derived stem cell injections. The different types of fistulas are drawn in red. According to Park’s classification: 1: superficial; 2: inter-sphincteric; 3: trans-sphincter; 4: supra-sphincteric; 5: extra-sphincteric. According to the American Gastroenterological Association: simple fistulas = 1 + 2; complex fistulas = 3 + 4 + 5. In blue, different ADSC routes that are generally applied in refractory complex fistulas: 1: via the internal orifice of the fistula; 2: via the external orifice of the fistula; 3: in the peri-fistula space.

**Figure 5 ijms-22-09967-f005:**
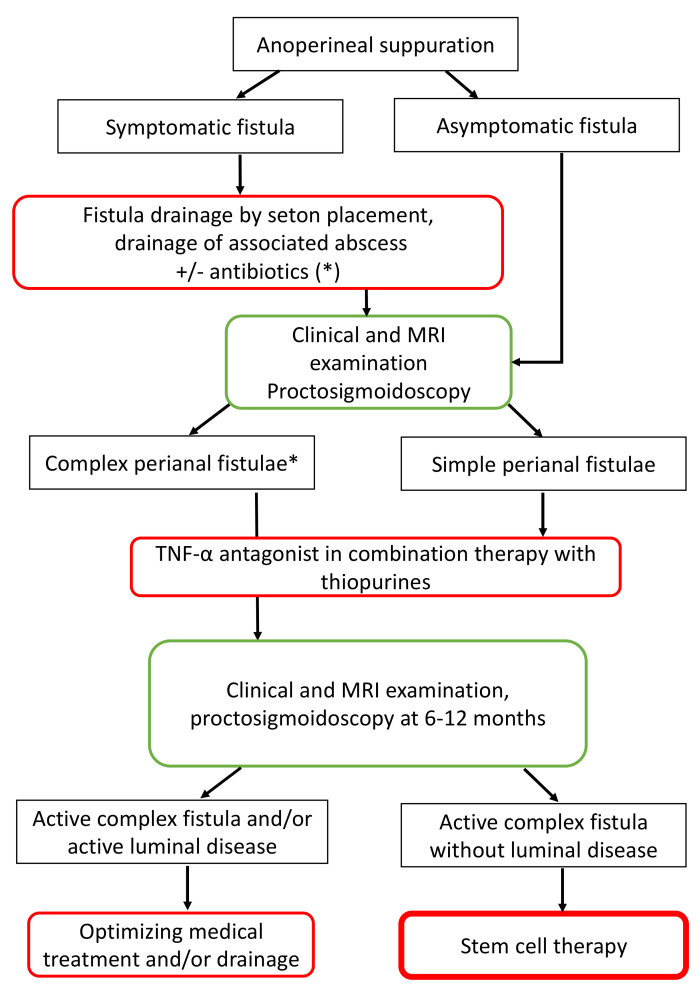
Algorithm for the management of perianal suppuration and complex fistula in Crohn’s disease including the place of stem cell therapy (*: ciprofloxacin and/or metronidazole).

**Table 1 ijms-22-09967-t001:** Main published studies that included patients with anoperineal fistula in Crohn’s disease who were injected in situ with adipose-derived stem cells.

Author, Year, [ref]	Phase, Design (Patient Number)	Cell Type (Additional Procedure)	Number of Cells Injected (×10^7^)	Concomitant Treatments	Evaluation Criteria	Time Point (Weeks)	Complete Healing	Total Follow-Up (Weeks)	Related Adverse Events
Garcia-Olmo 2009 [135]	II, prospective (49, 14 with CD)	Autologous (+fibrin glue vs. fibrin glue alone)	2	yes	Clinical + QoL	8	71%	48	none
Guadalajara, 2012 [136]	II, retrospective, monocentre (50)	Autologous? (+fibrin glue versus fibrin glue alone)	2	yes	Clinical + MRI	152	58%	168	1 (2%)
Lee, 2013 [137]	I, prospective, multicentre (43)	Autologous (+fibrin glue)	3	yes	Clinical	4, 6 and 8	79%	-	none
De La Portilla, 2013 [138]	I/IIa, multicentre (24)	Allogenic	2	yes	Clinical + MRI	24	56%	-	4 (16.5%)
Cho, 2015 [139]	II (43)	Autologous	3	yes	Clinical	48	79%	96	none
Panes, 2016 [67]	III, prospective, multicentre, controlled (212)	Allogenic (*)	12	yes	Clinical + MRI	24	50% versus 34% in placebo	24	18 (17%)
Panes, 2018 [68]	III, prospective, multicentre, controlled (212)	Allogenic (*)	12	yes	Clinical + MRI	52	56% versus 38% in placebo	52	-
Dietz, 2017 [140]	I, monocentre, open label single arm (12)	Autologous (+plug matrix)	2 (twice)	yes	Clinical	24	83%	24	none
Dige, 2019 [141]	I, monocentre, open label single arm (21)	Autologous SVF	-	yes		24	76%	24	1 (5%)
Serrero, 2019 [142]	I, monocentre, open label single arm (10)	Autologous SVF (+microfat grafting)	1 to 6	yes	Clinical + MRI	12 and 48	70 and 80% at 24 and 48 weeks respectively	48	3 (30%)
Herreros, 2019 [143]	Monocentre, observational (45 with 18 CD)	Allogenic & Autologous SVF (**)	0.2 to 6	no	Clinical	24	46%	26	none
Zhou, 2020 [144]	Monocentre, observational (22)	Autologous, versus surgery alone	0.5	yes	Clinical + MRI or EUS	12, 24 and 48	90, 72 and 63% at 12, 24 and 48 weeks respectively	48	none
Schwandner, 2021 [145]	Mononcenter Observational Retrospective (12)	Allogenic (*)	12	yes	Clinical	12	67%	56	4 (33%)

Studies that had included less than 10 patients have been excluded from this table. *: Cx601 (Davadstrocel, Alofisel^®^); **: multiple in situ injections in 5 cases. CD: Crohn’s disease; SVF = serum vascular fraction; QoL: quality of life; MRI: magnetic resonance imaging; EUS: endoscopic ultrasound. Related adverse events: adverse events in direct relation to ADSCs in situ injection.

**Table 2 ijms-22-09967-t002:** Main ongoing trials applying adipose-derived stem cells for Crohn’s disease anoperineal fistula.

ID Number Name Country	Phase (Number of Patients)	Protocol Design and Cells	Main Endpoints
NCT03466515 Stem Cell Treatment of Complex Crohn’s Perianal Fistula. A Pilot Study Denmark	I (20)	Injection in and around the fistula track of autologous fat tissue and ADSC from SVF (Cytori Celusion system^®^)	Healing (clinical + MRI), time to healing, AE
NCT03913572 Treatment of Perianal Disease Using Adipose-derived Stem Cells USA	Retrospective, observational (25)	Group 1: surgery + one single injection of ADSC derived from SVF (Revolve System—Acelity^®^) Group 2: surgery alone	Healing at 4 months Clinical + MRI Assessment of PDAI + Adverse Events
NCT03279081 Efficacy and Safety of Cx601, Adult Allogeneic Expanded ADSC for the Treatment of Complex Perianal Fistula(s) in Participants With Crohn’s Disease (CD) (ADMIRE-CD-II) USA	III Randomized, controlled (554)	Cx601 (Darvadstrocel, Alofisel) 12 × 10^7^ ADSC versus placebo, single intra-fistula injection	Clinical remission, clinical response, combined response (*) Clinical + MRI evaluation at 24 and 52 weeks TRAE
NCT04118088 Postauthorization Safety Study of the Long-Term Safety and Efficacy of Repeat Administration of Darvadstrocel in Patients With Crohn’s Disease and Complex Perianal Fistula EU, USA	IV (50)	Observational Repeat injection in patients 12 × 10^7^ ADSC (Cx601—Darvadstrocel—Alofisel)	Safety = TRAE
NCT04010526 Double-blind Randomised Placebo Controlled Study Evaluating Local Co-administration of Autologous Adipose-Derived Stromal Vascular Fraction with Microfat for Refractory Perianal Crohn’s Fistulas France	II (84)	Double-blind randomised Placebo Controlled Single injection of both ADSC from SVF and microfat tissue in the fistula track versus placebo injection	Healing at 24 weeks, clinical + MRI PDAI TRAE
NCT04075825 A Follow-up of a Phase 3 Study to Evaluate the Long-term Safety and Efficacy of Darvadstrocel in the Treatment of Complex Perianal Fistula in Subjects With Crohn’s Disease who have Participated in ADMIRE II Study USA	(≤554)	Participants who received a single dose of darvadstrocel (Cx601), 120 million cells, intralesionally or darvadstrocel matching placebo previously in the ADMIRE-CD II study will be observed for efficacy and safety	AE and TRAE from baseline to week 156 Clinical response and remission at weeks 104 and 156 Relapse(s)

*: clinical remission: closure of all treated external openings that were draining at baseline despite gentle finger compression; clinical response: closure of at least 50 percent (%) of all treated external openings that were draining at baseline despite gentle finger compression; combined response: closure of all treated external openings that were draining at baseline despite gentle finger compression, and absence of collections >2 cm at MRI; AE: adverse event; TRAE: treatment related adverse event; PDAI: Perianal Disease Activity Index; MRI: magnetic resonance imaging; SVF: serum vascular fraction; EU: European Union.

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
