# Peer review of "Adipose-Derived Stem Cells in the Treatment of Perianal Fistulas in Crohn’s Disease: Rationale, Clinical Results and Perspectives"

_ijms, 2021, doi:10.3390/ijms22189967_

Round 1

Reviewer 1 Report

In this manuscript by Buscail et al, the authors summarized recent trials and findings regarding the treatment of perianal fistulas in Crohn's disease (CD) using adipose-derived stem cells (ADSCs). This is a well-written review starting from introduction of CD and ADSCs, to summary of application of ADSCs in perianal fistula treatment. However, there are a number of minor concerns that need to be addressed before this manuscript is in a publishable fashion. Specific comments are as follows:

1. Section 3 is too long. It contains about 2/3 of the listed references but many are not directly related to the main subject or the content in later sections. It is suggested that section 3.3 and 3.6 may be shortened.

2. With the success in using ADSCs in treating CD fistulas, what are the molecular mechanisms behind these beneficial effects? As the authors described that there are preclinical CD models but none specific for CD fistulas, are there proposed or speculated mechanisms based on recent findings in preclinical or clinical models?

Author Response

In this manuscript by Buscail et al, the authors summarized recent trials and findings regarding the treatment of perianal fistulas in Crohn's disease (CD) using adipose-derived stem cells (ADSCs). This is a well-written review starting from introduction of CD and ADSCs, to summary of application of ADSCs in perianal fistula treatment. However, there are a number of minor concerns that need to be addressed before this manuscript is in a publishable fashion. Specific comments are as follows:

1. Section 3 is too long. It contains about 2/3 of the listed references but many are not directly related to the main subject or the content in later sections. It is suggested that section 3.3 and 3.6 may be shortened.

RESPONSE to reviewer 1: We fully agree with that. The section 3 has been shortened especially sub-section 3.3. In addition, the subsection 3.6 has been deleted and two sentences of this sub-section have been dispatched to the sub-section 3.5.

2. With the success in using ADSCs in treating CD fistulas, what are the molecular mechanisms behind these beneficial effects? As the authors described that there are preclinical CD models but none specific for CD fistulas, are there proposed or speculated mechanisms based on recent findings in preclinical or clinical models?

RESPONSE to reviewer 1: Thank you for this comment: the potential molecular mechanisms that could be attributed to the success of ADSCs in the treatment perianal fistula in CD have been now synthetized in the new sub-section 3.6 (second paragraph) with an additional new figure 3 and new reference (# 140).

All additions have been highlighted in yellow within the text. 

Reviewer 2 Report

The manuscript from Etienne Buscail et al. describes the possibility of using adipose-derived stem cells in the treatment of perianal fistulas in Crohn's disease: rationale, clinical results, and perspectives.

I read with interest the article, but the manuscript should be entirely revised since the written language presented does not allow a perfect understanding of the paper’s introduction and other sections.

Some selected examples:

„ CD are frequent, and sometimes inaugural” consider revising wording

 “searching for primary and secondary orifices” consider revising wording

Please explain the sentence: “anti-TNF alpha biotherapy”?

Please explain the sentence:  „non-negligible inflammatory state”

Behaviour? Reseachs?

Therefore, before a possible acceptance, the authors should deeply revise the manuscript in terms of language.

In addition, the authors should mention the association of adipose-derived stem cells with cancer progression.

Author Response

The manuscript from Etienne Buscail et al. describes the possibility of using adipose-derived stem cells in the treatment of perianal fistulas in Crohn's disease: rationale, clinical results, and perspectives. I read with interest the article, but the manuscript should be entirely revised since the written language presented does not allow a perfect understanding of the paper’s introduction and other sections.

Some selected examples:

„ CD are frequent, and sometimes inaugural” consider revising wording

 “searching for primary and secondary orifices” consider revising wording

Please explain the sentence: “anti-TNF alpha biotherapy”?

Please explain the sentence:  „non-negligible inflammatory state”

Behaviour? Reseachs?

Therefore, before a possible acceptance, the authors should deeply revise the manuscript in terms of language.

RESPONSE to reviewer 2: We fully agree with that and numerous sentences, expressions and grammar/synthax errors have been corrected throughout the text. Most of the corrections have been highlighted in yellow within the text. 

In addition, the authors should mention the association of adipose-derived stem cells with cancer progression.

REPONSE to reviewer 2: We agree and we have added this fact in the new sub-section 6 (firs paragraph) as well as new references (# 132 to 135)  including a recent ex vivo study that do not support a high risk of tumor cell proliferation. 

Reviewer 3 Report

Are there data concerning combination treatment of perianal disease combining biologic agents and Autologous adipose-derived stem cells?

In the conclusion part please emphasize that:

  • Local injections of Autologous ASCs following appropriate surgical preparation of a fistula, represents a promising combination strategy for management of resistant perianal fistulas.
  • At the moment there are no enough scientific data allowing the formation of unequivocal opinion regarding the use of stem cells in the treatment of perianal Crohn’s disease.
  • From a surgical point of view the fluid consistency of the preparation, results in incalculable losses of biological material during injection and low numbers of surviving cells being administered into the fistula.

Please refer to:

  • The bioavailability of stem cells, and the accompanying consequences in their survival.
  • The long-term clinical results (if they exist)
  • Mention the (few) experimental data concerning the efficacy of MSCs in intensification of healing through promotion of re-epithelialization, cell proliferation and enhanced angiogenesis.

Please provide the readers with a therapeutic algorithm for treatment of fistulas of Crohn’s disease indicating the time of stem cell treatment.

Author Response

Are there data concerning combination treatment of perianal disease combining biologic agents and Autologous adipose-derived stem cells?

RESPONSE to reviewer 3: in all studies the stem cells have been locally administered together with biotherapy and/or immunosuppressant.  This has been added in the sub-section 4.1

In the conclusion part please emphasize that: Local injections of Autologous ASCs following appropriate surgical preparation of a fistula, represents a promising combination strategy for management of resistant perianal fistulas. At the moment there are no enough scientific data allowing the formation of unequivocal opinion regarding the use of stem cells in the treatment of perianal Crohn’s disease.

RESPONSE to reviewer 3: These two sentences have been added in the last section (section 5) of the manuscript.

From a surgical point of view the fluid consistency of the preparation, results in incalculable losses of biological material during injection and low numbers of surviving cells being administered into the fistula. Please refer to: The bioavailability of stem cells, and the accompanying consequences in their survival.

RESPONSE to reviewer 3:  This point is important and we fully agree. It has been included in the perspectives (end of the section 5) together with the future study of biocompatibility between ADSCs and the transport/injection system.

The long-term clinical results

RESPONSE to reviewer 3: two sentences as well as a new reference (# 154) have been added at the end of sub-section 4.1.

Mention the (few) experimental data concerning the efficacy of MSCs in intensification of healing through promotion of re-epithelialization, cell proliferation and enhanced angiogenesis.

RESPONSE to reviewer 3: two sentences and references has been added at the end of the new subsection 3.6.

Please provide the readers with a therapeutic algorithm for treatment of fistulas of Crohn’s disease indicating the time of stem cell treatment.

RESPONSE to reviewer 3: Thank you for proposing that: an algorithm is presented in additional new figure 5 and comment in the subsection 5 together with 2 new references (# 159, 160) referring to published national and international recommendations.

All additions have been highlighted in yellow within the text.